# Human GBP1 Is Involved in the Repair of Damaged Phagosomes/Endolysosomes

**DOI:** 10.3390/ijms24119701

**Published:** 2023-06-02

**Authors:** Hellen Buijze, Volker Brinkmann, Robert Hurwitz, Anca Dorhoi, Stefan H. E. Kaufmann, Gang Pei

**Affiliations:** 1Department of Immunology, Max Planck Institute for Infection Biology, Charitéplatz 1, 10117 Berlin, Germany; 2Microscopy Core Facility, Max Planck Institute for Infection Biology, Charitéplatz 1, 10117 Berlin, Germany; 3Protein Purification Facility, Max Planck Institute for Infection Biology, Charitéplatz 1, 10117 Berlin, Germany; 4Institute of Immunology, Friedrich-Loeffler-Institut, Federal Research Institute for Animal Health, Greifswald-Insel Riems, 17493 Greifswald, Germany; 5Faculty of Mathematics and Natural Sciences, University of Greifswald, 17489 Greifswald, Germany; 6Emeritus Group of Systems Immunology, Max Planck Institute for Multidisciplinary Sciences, Am Fassberg 11, 37077 Göttingen, Germany; 7Hagler Institute for Advanced Study, Texas A&M University, College Station, TX 77843, USA

**Keywords:** guanylate-binding proteins, endolysosomal damage, *Mycobacterium tuberculosis*, *Listeria monocytogenes*

## Abstract

Mouse guanylate-binding proteins (mGBPs) are recruited to various invasive pathogens, thereby conferring cell-autonomous immunity against these pathogens. However, whether and how human GBPs (hGBPs) target *M. tuberculosis* (Mtb) and *L. monocytogenes* (Lm) remains unclear. Here, we describe hGBPs association with intracellular Mtb and Lm, which was dependent on the ability of bacteria to induce disruption of phagosomal membranes. hGBP1 formed puncta structures which were recruited to ruptured endolysosomes. Furthermore, both GTP-binding and isoprenylation of hGBP1 were required for its puncta formation. hGBP1 was required for the recovery of endolysosomal integrity. In vitro lipid-binding assays demonstrated direct binding of hGBP1 to PI4P. Upon endolysosomal damage, hGBP1 was targeted to PI4P and PI(3,4)P2-positive endolysosomes in cells. Finally, live-cell imaging demonstrated that hGBP1 was recruited to damaged endolysosomes, and consequently mediated endolysosomal repair. In summary, we uncover a novel interferon-inducible mechanism in which hGBP1 contributes to the repair of damaged phagosomes/endolysosomes.

## 1. Introduction

Guanylate-binding proteins (GBPs) constitute one of the most abundant GTPase families induced by type I and II interferons (IFNs) [1,2,3]. The regulation of GBPs expression by IFNs is highly conserved amongst vertebrates [4]. In addition to IFNs, several proinflammatory cytokines, such as TNF-α, IL-1α, and IL-1β induce expression of GBPs [5,6]. To date, seven human GBPs (hGBPs), eleven murine GBPs (mGBPs), and two mouse pseudogenes have been identified. Based on their structural and biochemical properties, GBPs are considered as the dynamin superfamily of GTPases [7]. Similar to dynamin, GBPs also possess an N-terminal globular GTP-binding domain (large G domain), a connecting middle domain, and a C-terminal GTPase effector domain (GED). A C-terminal CaaX motif is also present in GBP1, GBP2, and GBP5 of human and murine origin. This motif, which is also isoprenylated as in small Ras or Rab GTPases, mediates the targeting of GBPs to intracellular membranous compartments. Biochemically, GBPs have low affinity to GDP/GTP, but display high intrinsic hydrolysis of GTP to GMP via GDP. Both GTP-binding and hydrolysis of GBPs are essential for their oligomerization and association with membrane structures [8,9,10].

Upon induction, GBPs are delivered to various pathogen-containing vacuoles (PCVs). mGBP1, 3 and 10 colocalize with *Listeria monocytogenes* (Lm) and *Mycobacterium bovis* BCG soon after infection [11]. mGBP2 and mGBP2/5 are associated with *Salmonella typhimurium* [12] and *Francisella novicida* [13], respectively. Moreover, various mGBPs have been shown to colocalize with intracellular *Shigella flexneri*, *Burkholderia thailandensis*, *Legionella pneumophila*, *Yersinia pseudotuberculosis*, *Brucella abortus,* and *Chlamydia trachomatis* [14,15,16,17,18]. In addition, mGBPs are widely associated with intracellular parasites, including *Toxoplasma gondii* and *Leishmania major* [19,20,21,22]. Following their association with PCVs, GBPs further recruit different host factors to PCVs to activate distinct defense mechanisms. mGBP7 mediates the delivery of the NADPH oxidase complex to PCVs, contributing to oxidative killing of Lm and *M. bovis* BCG [11]. mGBP1 and mGBP7 recruit autophagy components and sequester pathogens in autolysosomes for degradation [11,12]. Furthermore, various GBPs promote activation of different types of inflammasomes [12,13,23,24,25,26]. Hierarchical recruitment of GBPs to cytosolic *Shigella flexneri* or *Burkholderia thailandensis* impairs actin-mediated motility, and thus restricts cell-to-cell spread of these pathogens [14,15,16]. Finally, GBPs display direct antimicrobial activities by attacking the plasma membrane of *Toxoplasma gondii* [22] and by destabilizing the outer membrane of Gram-negative bacterial envelopes via binding to LPS [27]. These studies emphasize that recruitment of GBPs to PCVs is critical for their host defense functions. However, most of these studies focused on mouse GBPs during infection with Gram-negative pathogens. Considering the diversification of GBPs between humans and mice, it is worth investigating whether and which human GBPs associate with PCVs.

Here, we show that hGBPs associate with intracellular *Mycobacterium tuberculosis* (Mtb) and Lm, but not with mutants that fail to destabilize the phagosomal membrane. hGBP1 specifically formed puncta structures upon endolysosomal damage. GTP-binding and isoprenylation of hGBP1 were critical for puncta formation. Functional analysis uncovered that hGBP1 mediated recovery of lysosomal integrity. hGBP1 directly bound to PI4P in vitro and endolysosomal damage stimulated the colocalization of hGBP1 with PI4P and PI(3,4)P2 in cells. Finally, live-cell imaging demonstrated that hGBP1 was recruited to damaged endolysosomes, followed by fusion with hGBP1-positive vesicles. Subsequently, the permeabilized membrane was sealed. Altogether, we uncovered a novel role of hGBP1 in which hGBP1 associates with permeabilized phagosomes or endolysosomes, and further contributes to the repair of damaged phagosomes/endolysosomes.

## 2. Results

### 2.1. Human GBP1/2 Are Recruited to Intracellular Mycobacterium tuberculosis and Listeria monocytogenes

We and others have previously demonstrated GBPs as critical members of biomarker signatures for active tuberculosis in various cohorts [28,29,30,31,32]. Whether and which hGBPs are recruited to Mtb or other Gram-positive pathogens remain unclear. To address these questions, THP-1 cells were infected with Mtb or Lm. Considering that THP-1 cells upon IFN-γ treatment only express hGBP1-5 [26], we thus focused on hGBP1-5 in our analysis. In contrast to Gram-negative pathogens which are fully decorated by GBPs, endogenous hGBP1-5 formed puncta structures which partially colocalized with Mtb (Figure 1A,B). However, no hGBPs puncta were recruited to *M. bovis* BCG (Figure 1A,B). Compared to Mtb, *M. bovis* BCG lacks the region of differentiation 1 (RD-1), which encodes virulence factors causing the rupture of phagosomes [33,34]. Therefore, we hypothesized that hGBPs were recruited to damaged phagosomes upon infection with pathogens able to perturb phagosomal membranes. To test this hypothesis, Lm, another intracellular Gram-positive pathogen that rapidly disrupts phagosomal membranes [35], was employed to infect cells expressing various hGBPs. Indeed, hGBP1 and hGBP2 puncta colocalized with Lm (Figure 1C), yet recruitment of other hGBPs was not found even after IFN-γ treatment (Figure 1D, Appendix AA). Listeriolysin O (LLO) is crucial for the phagosomal membrane perturbation by Lm, allowing for its escape from phagosomes into the cytosol [36]. To further validate our notion, Listeria LLO-deficient mutant (Listeria ∆hly) was employed. Similar to *M. bovis* BCG, hGBP1 puncta was not recruited to Listeria ∆hly mutant that remained within intact phagosomes (Figure 1E,F). Therefore, we conclude that hGBP1/2 puncta associate with Mtb and Lm, but not with mutants that fail to cause rupture of phagosomes.

### 2.2. hGBP1 Forms Puncta Structures upon Endolysosomal Damage

It has been shown that the insertion of type III or type IV secretion systems from *Yersinia* and *Legionella* into PCVs triggers the recruitment of GBPs [17]. Mtb and Lm substantially differ from the mechanisms of phagosome perturbation induced by the above-mentioned Gram-negative pathogens. Accordingly, it appears unlikely that hGBPs recognize conserved components from diverse Gram-positive and -negative pathogens. We hypothesized that hGBPs may sense the integrity and/or disruption of endocytic/phagocytic compartments. To interrogate our assumption, cells expressing individual hGBPs were stimulated with the lysosomotropic compound L-Leucyl-L-leucine methyl ester (LLOMe). LLOMe is polymerized by the thiol protease dipeptidyl peptidase I, leading to complete permeabilization of endolysosomes [37]. Consistent with a previous report [17], the formation of mCherry-hGBP1 puncta, but not other hGBPs, in the perinuclear region was significantly induced by LLOMe stimulation (Figure 2A,B). Staining for endogenous hGBP1 confirmed the formation of hGBP1 puncta upon LLOMe stimulation (Figure 2C). The expression of hGBP1 was detected under the basal condition and its expression level was drastically induced by IFN-γ (Appendix AB). Furthermore, mCherry-hGBP1 puncta were markedly induced by IFN-γ treatment, indicating other IFN-γ inducible factors possibly contributing to hGBP1 puncta formation upon endolysosomal damage (Figure 2D,E). Finally, to determine whether hGBP1 puncta formation is specific to this damage, cells were exposed to various reagents inducing the disruption of plasma membrane. Phalloidin-positive staining indicated plasma membrane perforation by digitonin and LLO; however, no clear hGBPs puncta were observed (Figure 2F,G). Altogether, we conclude that hGBP1 puncta are specifically induced by endolysosomal damage.

### 2.3. GTP-Binding and Isoprenylation of hGBP1 Are Critical for Its Puncta Formation

To elucidate the mechanism of hGBP1 puncta formation upon lysosomal damage, hGBP1 knockout (KO) cells were generated using Cas9-CRISPR technology. Western blotting confirmed the complete loss of hGBP1 (Figure 3A). hGBPs puncta were diminished in hGBP1 KO cells, supporting a critical role of hGBP1 upon endolysosomal damage (Figure 3B). To understand the molecular determinants essential for hGBP1 puncta formation upon endolysosomal damage, various hGBP1 mutants were generated and expressed in hGBP1 KO cells. Mutations in the GTPase domain (K51A and S52N), which are nucleotide free [38], completely abolished formation of hGBP1 puncta (Figure 3C,D). Interestingly, the GTP-binding, but GTPase-deficient mutant-hGBP1 R48A [38] was still able to form puncta structures (Figure 3C,D). Furthermore, the contribution of the CaaX-box and a polybasic motif at the C-terminus to hGBP1 puncta formation were examined. Isoprenylation of the CaaX-box is essential for hGBPs anchoring to intracellular membranes [39]. Consistently, deletion of the CaaX-box completely abrogated hGBP1 puncta formation, suggesting association of hGBP1 with membranous structures upon endolysosomal damage (Figure 3C,D). While targeting of GBP1 to cytosolic bacteria depends on the C-terminal polybasic motif [14], the mutant of this motif (PBmut, RRRK_584–587_AAAA) still displayed puncta formation. The R227E/K228E double mutation induces constitutive dimer formation and localizes in vesicles [39]. Indeed, we observed association of the R227E/K228E mutant with small vesicles at basal conditions. Upon endolysosomal damage, this mutant formed larger puncta structures compared to hGBP1 WT (Figure 3C,D). Altogether, these results indicate that GTP-binding and isoprenylation rather than GTP hydrolysis of hGBP1 are essential for hGBP1 puncta formation in response to endolysosomal damage.

**Figure 2 ijms-24-09701-f002:**
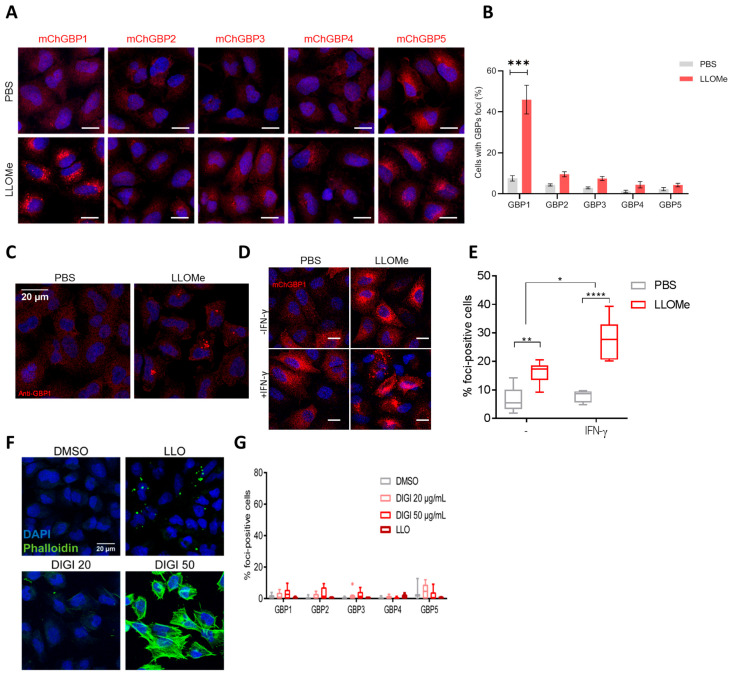
Endolysosomal damage induces the formation of hGBP1 foci. (**A**) Representative images of A549 cells expressing various mCherry-hGBPs upon lysosomal damage. (**B**) Quantification of the percentage of GBPs foci-positive cells in (**A**). (**C**) Endogenous hGBP1 staining upon lysosomal damage. (**D**) Representative images of A549 cells expressing mCherry-hGBP1 with or without IFN-γ priming upon lysosomal damage. (**E**) Quantification of the percentage of cells with hGBP1 foci in (**D**). (**A**–**E**) cells were treated with mock or LLOMe (1 mM) for 1 h to induce lysosomal damage. (**F**) Representative images of Phalloidin staining upon plasma membrane damage. (**G**) Quantification of the percentage of cells with hGBPs foci upon plasma membrane damage. (**F**,**G**) A549 cells expressing various mCherry-hGBPs stimulated with DMSO, digitonin (DIGI, 20 or 50 µg/mL), or Listeriolysin O (LLO). Scale bar: 20 µm. Data are shown as mean ± SD of three independent experiments; *p*-values were calculated using two-way ANOVA with Bonferroni’s multiple comparison test. (*) *p* < 0.05, (**) *p* < 0.01, (***) *p* < 0.001, (****) *p* < 0.0001.

### 2.4. hGBP1 Mediates Recovery of Endolysosomal Integrity

To understand the roles of hGBP1 in endolysosomal damage, the intracellular distribution of hGBP1 was investigated upon LLOMe treatment. hGBP1 partially colocalized with Rab5 or LAMP2, which are the markers for early endosomes or late endosomes/lysosomes, indicating the association of hGBP1 to damaged endolysosomal vesicles (Appendix AA,B). It has been demonstrated that the endosomal sorting complex required for transport (ESCRT)-III machinery is recruited to damaged endolysosomes and mediates their repair [40,41]. Indeed, hGBP1 colocalized with ESCRT-III proteins-CHMP2A and CHMP6 upon LLOMe treatment (Appendix AC,D). Therefore, we hypothesized that hGBP1 likely contributes to the repair of disrupted endolysosomes. To examine this hypothesis, we employed Lysotracker Red (LTR) to evaluate the integrity of lysosomes in response to LLOMe treatment. LTR becomes protonated upon entry into lysosomes, and thus selectively accumulates in lysosomes. Upon lysosomal permeabilization, LTR staining fails to remain in lysosomes due to proton leakage [42]. In agreement with previous reports [43,44,45], the signal of LTR staining was diminished after only 30 min of LLOMe treatment, suggesting the loss of lysosomal integrity. The signal of LTR gradually increased from 60 min, indicating the recovery of endolysosomal integrity. Consistent with a previous report [45], at 240 min post LLOMe treatment, endolysosomes fully recovered as shown by the LTR staining (Figure 4A,B). Interestingly, compared to the untreated control, both abundance and intensity of LTR-positive vesicles were higher at 240 min after LLOMe treatment, suggesting that endolysosomes recover to values exceeding those before damage. Moreover, the size of regenerated endolysosomes was elevated (Figure 4A,B). To investigate the potential role of hGBP1 in this process, we employed hGBP1 KO cells to monitor the recovery of LTR staining upon LLOMe. While both of the hGBP1 KO clones displayed comparable loss of LTR staining with the control cells after 30 min of LLOMe treatment, the recovery of LTR staining after 2 and 4 h of LLOMe treatment was significantly impaired in hGBP1 KO cells (Figure 4C,D). However, this impairment by hGBP1 KO was counteracted by IFN-γ, indicating that other IFN-γ inducible factors are also involved in the recovery of lysosomal damage. Altogether, hGBP1 along with other IFN-γ inducible factors are required for restoring the integrity of endolysosomes upon their membrane permeabilization.

### 2.5. hGBP1 Is Targeted to Damaged Endolysosomes Likely via Binding to the Membrane Lipids, PI4P and PI(3,4)P2

Dynamin associates with membranes via directly binding to phosphatidyinositol-4,5-bisphosphate (PIP2) [46]. We hypothesized that hGBPs, as members of the dynamin GTPase superfamily, could directly bind to membrane lipids. To this end, mCherry-hGBP1-5 with 6xHis tag were expressed in *E. coli*, purified (Figure 5A) and tested for their abilities to bind to membrane lipids in vitro. While hGBP2 and hGBP5 did not show any binding to membrane lipids, hGBP1, hGBP3, and hGBP4 were differentially bound to distinct lipids (Figure 5B). hGBP1 showed strong interaction with the plasma membrane lipid, phosphatidylserine (PS), and modest interaction with phosphatidylinositol 4-phosphate (PI4P) and phosphatidylcholine (PC). hGBP3 displayed strong binding to PI4P and moderate binding to PC, while hGBP4 was solely bound to PS. To further evaluate the binding of hGBP1 to phosphatidylinositol lipids in cells, we employed several GFP-tagged lipid sensors (PH-PLCD1 for PI(4,5)P2, PH-Btk for PIP3, PH-Akt for PIP3/PI(3,4)P2, P4M-SidM for PI4P) [47,48] to investigate their colocalization with hGBP1 upon lysosomal damage. PI4P is present in cis-Golgi complex, trans-Golgi network (TGN), plasma membrane, secretory vesicles, and late endosomes/lysosomes [49]. PIP2 and PIP3 are mainly localized on plasma membranes [50,51,52], and PI(3,4)P2 predominately on lysosomes [51,52,53]. Upon mock treatment, hGBP1 puncta and their colocalization with the lipids tested were not observed (Figure 5C). Furthermore, no PI(3,4)P2-positive vesicles were found in mock-treated cells. In contrast, LLOMe treatment induced the formation of distinct PI4P and PI(3,4)P2-positive vesicles. Moreover, hGBP1 accumulated in the perinuclear region and colocalized with PI4P or PI(3,4)P2 on vesicles (Figure 5C), resembling the Golgi association of hGBP1 reported earlier [10]. To substantiate the Golgi association of hGBP1, cells were transfected with galactosyltransferase (GalT)-GFP to visualize the Golgi complex. No clear Golgi association of hGBP1 was observed under basal condition; however, hGBP1 clearly colocalized with GalT upon LLOMe treatment, confirming its Golgi association (Appendix AE). Despite hGBP1 binding to plasma membrane lipid-PS in vitro, no colocalization of hGBP1 with PIP2 or PIP3 was observed in cells (Figure 5C). Finally, by using a specific PI4P antibody, we found colocalization of hGBP1 puncta with endogenous PI4P upon endolysosomal damage (Figure 5D). Altogether, upon endolysosomal damage, hGBP1 is associated with the Golgi complex and late endosomes/lysosomes, likely by directly binding to PI4P and PI(3,4)P2.

### 2.6. hGBP1 Contributes to the Repair of Damaged Endolysosomes

PI4P is critical for maintaining lysosomal identity by regulating lysosomal efflux and facilitating efficient sorting of lysosomal contents [54]. It also controls the trafficking from TGN to lysosomes [55]. Therefore, we hypothesized that PI4P could mediate the targeting of hGBP1 from Golgi complex to damaged lysosomes. Indeed, the majority of PI4P colocalized with LAMP2 and hGBP1 colocalized with both LAMP2 and PI4P upon endolysosomal damage (Figure 6A). To elucidate molecular determinants of hGBP1 essential for its colocalization with LAMP2, various hGBP1 mutant-expressing cells were treated with LLOMe, and subsequently stained against LAMP2. hGBP1 K51A, S52N, and ΔCaaX mutants failed to colocalize with LAMP2, suggesting that GTP-binding and isoprenylation are critical for its targeting to damaged endolysosomes (Appendix AA). Galectin-3/8 recognize damaged endosomes/lysosomes via binding to β-galactoside carbohydrates in the luminal side of endosomes/lysosomes [56,57]. To investigate the dynamics of hGBP1 recruitment to damaged endolysosomes, live-cell imaging was performed with cells expressing mCherry-hGBP1 and YFP-Galectin-3. Shortly after endolysosomal damage, Galectin-3 was recruited to damaged endolysosomes. After approximately 20 min, hGBP1 was recruited and colocalized with Galectin-3 (Figure 6B). Similarly, upon Lm infection, hGBP1 colocalization with endogenous Galectin-8 on the bacteria was also observed (Appendix AB). Additionally, hGBP1-positive vesicles were found to fuse with damaged endolysosomes (Figure 6C). The signal of Galectin-3 diminished over time, while hGBP1 was still preserved on these endolysosomes (Figure 6C,D). This could be due to the sequestration of damaged endolysosomes by autophagosomes [58]. Accordingly, we observed the colocalization of hGBP1 with LC3B upon endolysosomal damage (Appendix AF). Yet, whether and how hGBP1 is involved in autophagy-mediated sequestration of damaged lysosomes require further investigations. Altogether, we conclude that hGBP1 is recruited to damaged endolysosomes and likely contributes to their repair.

### 2.7. hGBP1 Does Not Affect Intracellular Survival of Mtb and Lm

Since hGBP1 mediates the repair of damaged endolysosomes, we speculated that hGBP1 contributes to the control of bacterial infections. To this end, THP-1 non-targeting and hGBP1 KO cells were infected with Mtb or Lm for various time points and their intracellular survival was investigated. PMA differentiation alone induced the expression of hGBP1, hGBP2, and hGBP3 (Appendix AC), and thus we did not utilize IFN-γ to stimulate hGBPs expression in our experiment. The intracellular growth of Mtb (Appendix AA) or Lm (Appendix AB) was not significantly affected by hGBP1 KO. To exclude the possibility that other hGBP members compensate hGBP1 KO, A549 cells were transfected with mock or individual hGBPs, and thereafter infected with Lm. Consistent with hGBP1 KO, overexpression of hGBP1 or other individual hGBPs did not restrict intracellular growth of Lm (Appendix AC). Altogether, hGBP1 alone does not contribute to the control of intracellular growth of Mtb and Lm. 

## 3. Discussion

Elucidation of the mechanism by which GBPs are recruited to intracellular pathogens is crucial for a mechanistic understanding of the cell-autonomous immunity conferred by GBPs. Previously, it was shown that IRGMs dissociation from PCVs leads to mGBP2 recruitment to vacuoles containing *Chlamydia trachomatis* and *Toxoplasma gondii* [59]. Ubiquitin association also promotes mGBPs delivery into PCVs [60]. Moreover, Galectin-3 is known to mediate the delivery of mGBPs to vacuoles that are perturbed by type IV or III secretion system from legionellae or yersiniae, respectively [17]. It has been proposed that GBPs target vacuoles containing *Toxoplasma* or *Salmonella* and facilitate the disruption of those vacuolar membranes [12,59,61,62,63]. However, how hGBPs are delivered into vacuoles harboring Mtb and Lm remains unclear. In this study, we employed Mtb and Lm which both induce perturbation of the phagosomal membranes via different mechanisms. Mtb secrets ESAT-6 via ESX-1 type VII secretion system to perforate the phagosomal membrane, while Lm utilizes the pore-forming cholesterol-dependent toxin LLO to efficiently escape into the cytosol [33,36]. Upon Mtb and Lm infection, hGBP1/2 form puncta structures and partially colocalize with these two pathogens. In contrast, Gram-negative pathogens, such as *salmonellae*, *legionellae,* and *yersiniae*, are fully furnished with GBPs. This discrepancy could be due to direct LPS-binding by GBPs during infection with Gram-negative pathogens [26,27,64]. The colocalization of hGBPs with Mtb or Lm is dependent on ESX-1 or LLO, respectively, suggesting that damage of the phagosomal membrane triggers the association of hGBPs with Mtb and Lm. Consistently, LLOMe, which efficiently causes permeabilization of endolysosomes, also induced targeting of hGBP1 puncta to damaged endolysosomes. However, the damage of plasma membrane does not result in the same effect. Therefore, hGBP1 specifically targets the damaged phagosomes or endolysosomes and contributes to their repair, rather than destabilizing the phagosomal membrane containing Mtb or Lm. Galectin-3 has been shown to recruit mGBP2 onto damaged lysosomes [17]. Whether the recruitment of hGBP1 to damaged phagosomes/lysosomes is mediated by Galectin-3 needs to be further investigated.

Dynamin directly binds to PI(4,5)P2, thereby leading to membrane targeting [46]. For the first time, we uncovered that hGBP1/3/4, members of the dynamin GTPase superfamily, also directly bind to PI4P, PS, and other lipids in vitro. Upon lysosomal damage, hGBP1 colocalized with PI4P or PI(3,4)P2 on late endosomes/lysosomes. Therefore, similar to dynamin, hGBP1 could be targeted to late endosomes/lysosomes via directly binding to PI4P or PI(3,4)P2. In contrast to dynamin, hGBP1 lacks the PH domain which mediates lipid-binding [46]. We notice that the C-terminal polybasic motifs are only present in hGBP1/3. It is possible that the C-terminal polybasic motif of hGBP1 facilitates its binding to lipids. Consistently, this polybasic motif of hGBP1 has been shown to be critical for its association with cytosolic *S. flexneri* [65]. However, the precise mechanisms of how hGBP1 binds to PI4P and PI(3,4)P2 need to be elucidated.

Upon lysosomal damage, cells initiate ESCRT-mediated repair of lysosomal membranes and removal of damaged lysosomes by autophagy to restore lysosomal functionalities. Shortly after LLOMe treatment, ESCRT components are transiently recruited to damaged lysosomes (peaking at 10 min) in a TSG101- and ALIX-dependent manner [40,41]. ESCRT quickly repairs small ruptures of lysosomal membranes via forming filaments to directly seal the damage [40]. ESCRT is also recruited to mycobacteria-containing phagosomes in response to phagosome damage [66,67]. Moreover, mycobacteria-induced phagosomal disruption can activate LRRK2, which accordingly mediates Rab8A association. Consequently, LRRK2 and Rab8A together coordinate ESCRT recruitment to damaged endolysosomes [68]. The association of Galectin-3 with damaged lysosomes occurs significantly later and lasts for several hours [58]. Furthermore, the ubiquitin coating and autophagy machineries are recruited to damaged lysosomes in a process dependent on Galectin-3 and ultimately they are engulfed by autophagosomes [44,69,70]. Our study revealed that hGBP1 recruitment to damaged endolysosomes occurred with a delay, approximately 20 min after Galectin-3 association. In contrast to the early and transient association of ESCRT, hGBP1 association lasted for several hours, even after full repair of the damage indicated by the disappearance of Galectin-3. The detailed mechanism by which hGBP1 mediated the repair of damaged endolysosomes, especially how hGBP1 is coordinated with Galectin-3, ESCRT, and LRRK2 in this process, needs to be elucidated.

Numerous bacterial pathogens employ distinct mechanisms to rapidly induce the rupture of phagosomes and translocate into cytosol [71]. In contrast, host immune surveillance pathways can recognize cytosolic bacteria and activate type I IFNs (IFN-I). It is conceivable that the rupture of phagosomal membranes induced by bacteria is more extensive. Therefore, ESCRT may not suffice for the repair of damaged phagosomal membranes. Upon IFN-I activation, hGBP1 is markedly induced and recruited to disrupted phagosomes, consequently mediating the damaged phagosomes possibly via promoting fusion with hGBP1-positive vesicles. Therefore, hGBP1-mediated pathway may represent a novel IFN-inducible mechanism counteracting damaged phagosomes. However, hGBP1 alone did not play a significant role in the control of intracellular replication of Mtb or Lm. In line with our observation, mouse GBPs chromosome 3 KO does not affect Mtb intracellular replication [72], which could be due to the invasion mechanisms, by which Mtb and Lm evade GBPs-mediated host defense.

## 4. Material and Methods

### 4.1. Cell and Bacteria Culture

The human monocytic cell line THP-1 was obtained from the American Type Culture Collection (ATCC, TIB-202) and maintained in RPMI 1640 (Gibco, 31870, New York, NY, USA) with 10% (v:v) heat-inactivated fetal bovine serum (Sigma-Aldrich, F0804, St. Louis, MO, USA), 1 mM sodium pyruvate (Gibco, 11360070), 2 mM L-glutamine (Gibco, 25030081), 10 mM HEPES buffer (Gibco, 15630080), pH 7.2–7.5, 50 µM 2-mercaptoethanol (Gibco, 31350010). To differentiate THP-1 into macrophage-like cells, THP-1 cells were stimulated with 50 ng/mL phorbol 12-myristate 13-acetate (Sigma-Aldrich, P8139) for 72 h, and then incubated with fresh medium for another 72 h. A549 cells (ATCC, CCL-185, Manassas, VA, USA) and HEK293T cells (ATCC, CRL-11268) were maintained in complete Dulbecco’s modified Eagle’s medium (DMEM, Gibco), 4.5 g/L glucose (Gibco, 10938) with 10% (v:v) heat-inactivated fetal bovine serum, 1 mM sodium pyruvate, 2 mM L-glutamine. All cells were cultured in a humidified atmosphere at 37 °C and 5% CO_2_. All generated cell line stocks were tested and found to be mycoplasma-free.

*M. tuberculosis* H37Rv and *M. bovis* BCG were grown in 7H9 medium (Sigma-Aldrich, M0178) containing 0.05% Tween-80 (Sigma-Aldrich, P8074) to OD_600_ less than 0.6. Single bacteria were prepared by passing through syringes. Then, cells were infected with bacteria at a multiplicity of infection (MOI) of 10.

*L. monocytogenes* EGD WT and *L. monocytogenes* EGD ∆hly were kindly provided by Dr. Marc Lecuit (Institute of Pasteur, Paris, France). Cells were grown in brain heart infusion (BHI; Sigma-Aldrich, 53286) medium overnight to stationary phase, and then subcultured 1:10 in BHI medium for 2 h at 37 °C. Bacteria were then washed three times in DMEM medium. Cells were infected with bacteria at MOI of 5.

### 4.2. Plasmids, Antibodies, and Reagents

hGBP1, hGBP3, and hGBP4 were acquired from Dharmacon, UK. hGBP2 and hGBP5 ORF plasmids were obtained from Sino Biological, China. PH-PLCD1-GFP (#51407), PH-AKT-GFP (#51465), PH-Btk-GFP (#51463), GFP-P4M-SidMx2 (#51472) were obtained from Addgene, Watertown, MA, USA. YFP-Galectin-3 was kindly provided by Dr. Felix Randow (MRC Laboratory of Molecular Biology, Cambridge, UK).

L-Leucyl-L-Leucine methyl ester (LLOMe) (Cayman Chemicals, 16008, Ann Arbor, MI, USA), Digitonin (Merck, 300410, Rahway, NJ, USA), Listeriolysin O (Raybiotech, 228-11051-2, Peachtree Corners, GA, USA), recombinant human IFN-γ (PeproTech, 300-02, Cranbury, NJ, USA), Phalloidin Alexa Fluor™ 488 (ThermoFisher, A12379, Waltham, MA, USA), Lysotracker Red DND-99 (ThermoFisher, L7528) were purchased as indicated.

Antibodies used in this study include: Anti-ACTB (Sigma-Aldrich, A2228), anti-Cathepsin D (Cell Signaling Technology, 22845, Danvers, MA, USA), anti-EEA1 (BD Biosciences, 610456, San Jose, CA, USA), anti-LAMP2 (Santa Cruz Biotechnology, SC-18822, Dallas, TX, USA), anti-GBP1 (Santa Cruz Biotechnology, SC-53857), anti-GBP1-5 (Santa Cruz Biotechnology, SC-166960), anti-Galectin-8 (Abcam, ab109519, Cambridge, UK), anti-Listeria (ThermoFisher, PA1-30487), anti-mCherry (ThermoFisher, PA5-34974), anti-GFP (Proteintech, 66002-1-Ig, San Diego, CA, USA), anti-HA tag (Cell Signaling Technology, 3724S), anti-PI(4)P IgM (Echelon Biosciences, Z-P004), and secondary antibodies conjugated with different fluorophores (ThermoFisher).

### 4.3. mCherry-hGBPs Lentiviral Vector Cloning

mCherry-hGBPs were cloned using Gateway technology. hGBP ORFs were amplified using the Phusion high-fidelity polymerase. mCherry and hGBPs were cloned into the Gateway entry vector pENTR4 using a triple ligation reaction. mCherry-hGBP1-5 were shuttled to the lentiviral pLenti6/v5DEST vector by recombination using Gateway clonase. Recombination reaction products were transformed into *E. coli* Stbl3 competent cells and the right insertions were verified by sequencing. The respective PCR primers for GBPs cloning are described below in Table 1.

### 4.4. hGBP1 Mutagenesis

hGBP1 mutants were generated using Q5-site-directed mutagenesis kit. All hGBP1 mutations were verified by sequencing. Primers for mutagenesis are listed in Table 2.

### 4.5. Lentivirus Production

One day prior to transfection, HEK293T cells were split and seeded into a 6-well plate. For each well of a 6-well plate, 2.2 μg of pLenti6/v5-DEST Gateway vectors containing individual mCherry-GBPs were transfected along with 1.15 μg pLP1, 0.58 μg pLP2, and 0.88 μg pLP-VSV-G plasmid. The plasmid mixture was diluted in 300 μL serum- and antibiotic-free Opti-MEM medium, and incubated for 20 min with a premix of 12 μL Lipofectamine 2000 and 300 μL Opti-MEM. Cells were washed and covered in 1 mL Opti-MEM per well. The DNA-Lipofectamine mix was added at 600 μL per well and cells were incubated for 4 h before the addition of 1.5 mL full growth medium supplemented with antibiotic-antimycotic. Then, the medium was changed with high serum (20% FBS) complete growth medium 24 h after transfection. The medium containing different viruses was harvested at days 2 and 3 after transfection. The supernatants were filtered through a 0.45 μm filter and used for lentiviral transduction or stored at −80 °C.

### 4.6. hGBP1 KO Using CRISPR/Cas9 Technology

Oligos targeting hGBP1 were chosen from the human GeCKOv2 library and analyzed with CHOPCHOP to exclude off-targets or far C-terminal targeting. The targeting sequences against hGBP1 are: 5′-TTTAGTGTGAGACTGCACCG-3′ (sgRNA1) and 5′-GTGCCCCACCCCAAGAAGCC-3′ (sgRNA2).

The targeting sequences were synthesized as single-stranded oligos with flanking sequences for ligase-independent cloning (LIC) (GGAAAGGACGAAACACCG[targeting sequences]GTTTTAGAGCTAGAAATAGCAAGTTAAAATAAGG). The Cas9-containing plasmid U6-gRNA-CMV-EGFP-2A-CAS9-BHGpA was digested with ApaI and SpeI and the cleaved vector was treated with T4 DNA polymerase, and subsequently annealed with a short universal reverse strand oligo (5′AACGGACTAGCCTTATTTTAACTTGCTATTTCTAGCTCTAAAAC3′) and hGBP1 targeting oligos. LIC-annealed vectors were transformed into *E. coli* TOP10 and the plasmids were verified by Sanger sequencing. To generate lentiviral CRISPR vectors, hGBP1 sgRNAs were cloned into lentiCRISPR V2 vector and lentiviruses were produced by co-transfection with psPAX2 and pMD2.G as previously described [73].

A549 cells were nucleofected with U6-gRNA-CMV-EGFP-2A-CAS9-BHGpA plasmids. The transfected cells were cultured for 3 days and Cas9-expressing cells were sorted by FACS for GFP-positive cells. To generate THP-1 hGBP1 KO, cells were transduced with lentivirus harboring sgRNAs and selected with puromycin after 3 days of transduction. Single cells were seeded into 96-well plates by limiting dilutions. Positive KO clones with out-of-frame indels were selected by using RNAseq of barcoded amplicons.

### 4.7. Western Blot

Cells were washed two times with ice-cold DPBS and lysed in RIPA buffer (25 mM Tris-HCl pH 7.6, 150 mM NaCl, 1% NP-40, 1% sodium deoxycholate, 0.1% SDS) for 30 min on ice. Cell debris was spun down for 10 min at 16,000× *g* and 4 °C, and protein lysate was collected and boiled in reducing sample buffer for 10 min at 95 °C. Proteins were separated on 4–15% SDS gels (Bio-Rad, 4561086, Hercules, CA, USA) and transferred onto nitrocellulose membranes. Membranes were incubated with primary antibodies at 4 °C overnight and secondary antibodies at RT for 1 h, respectively. All primary antibodies were diluted in PBST (PBS supplemented with 0.1% Tween-20 (Sigma-Aldrich, P1379) containing 5% BSA. Membranes were developed with ECL detection kit (ThermoFisher Scientific, 34094) and imaged with the ChemiDoc system (Bio-Rad Laboratories). The antibody against ACTB/β-actin was used as the loading control.

### 4.8. mCherry-hGBPs Protein Purification

mCherry-hGBP1-5 proteins and mCherry control were expressed using pet28a vector with 6xHis-tag. Transformed *E. coli* BL21 RILL were grown in LB broth with 50 μg/mL kanamycin overnight at 37 °C, 220 rpm. Fresh LB with kanamycin was inoculated with overnight culture at 1:100, and grown to OD_600_~0.8. Protein expression was induced by the addition of 1 mM IPTG. Bacterial pellets were harvested after 6 h of induction at 37 °C. Proteins were purified by standard protocols of nickel affinity chromatography by the Core Facility for Protein Purification at the Max-Planck Institute for Infection Biology in Berlin, Germany.

### 4.9. In Vitro Lipid-Binding Assay of hGBPs

Membrane lipid strips (Echelon Biosciences, P-6002) were blocked in 3% BSA/PBST for 1 h by gently shaking at RT. Purified mCherry-hGBP1-5 proteins or GST-PLC-d1-PH as the positive control were added at 2 μg/mL in GTP-containing binding buffer (3% BSA, 5 mM MgCl_2_, 100 μM GTP/PBST) and incubated by gently shaking for 4 h at RT. Lipid strips were washed 3× with 10 mL PBST, and incubated with primary antibody against mCherry or GST in blocking buffer by gently shaking for 1 h at RT. The washing step was repeated and strips were incubated with secondary HRP-tagged antibody in blocking buffer for 1 h at RT. Membranes were washed as described above, developed with ECL detection kit, and imaged with the ChemiDoc system.

### 4.10. LysoTracker Red Assay

For LysoTracker retention assays, cells were pre-incubated with 75 nM LysoTracker Red in complete growth medium for 30 min before the addition of LLOMe (1 mM) in LysoTracker-containing medium. Cells were fixed at the indicated time points and images were taken using an automatic image acquisition system, the Arrayscan XTI platform.

### 4.11. Confocal Microscopy, Automatic Image Acquisition, and Image Analysis

For confocal microscopy, cells were grown on coverslips in 24-well plates and fixed with 4% formaldehyde/PBS for 10 min. Mtb-infected cells were fixed in 4% formaldehyde/PBS for 30 min, followed by overnight fixation in 1% formaldehyde/PBS. Cells on coverslips were quenched with 50 mM glycin for 10 min and permeabilized with 0.05% saponin with 1% BSA for 10 min. Cells were washed two times for 5 min with PBS and incubated with primary antibody at 4 °C overnight. After three PBS washes of 5 min each, secondary antibody was added for 1 h at RT. After washing as described above, cells were mounted with a DAPI-containing mounting medium and imaged with Leica SP8 confocal microscope with 40× magnification. 

For automatic image acquisition, cells were grown in 96-well plates and stimulated as indicated. Thereafter, cells were fixed with 4% formaldehyde/PBS for 10 min. DAPI (Sigma-Aldrich, D9542) was used for nuclear staining. Then, Arrayscan XTI platform was employed to image all wells of the plates.

Arrayscan images were analyzed using the High-Content Analysis (HCS) Studio Cell Analysis Software 4.0. Spot detection and quantification of fluorescence intensity and size were performed with CellProfiler software (https://cellprofiler.org/). Cells with high overexpression of mCherry were excluded from the analysis. GBPs foci were defined based on the background threshold method for each cell individually, using a stringent threshold correction factor adjusted individually for each experiment. Only objects of a minimum diameter of 0.71 μm were defined as spots.

### 4.12. Statistical Analysis

Statistical analysis was performed with GraphPad Prism v6.04 (GraphPad software Inc., San Diego, CA, USA). The *p*-values were calculated using Student’s *t*-test, one-way, or two-way ANOVA (Bonferroni’s multiple comparison test), and 95% confidence interval.

## Figures and Tables

**Figure 1 ijms-24-09701-f001:**
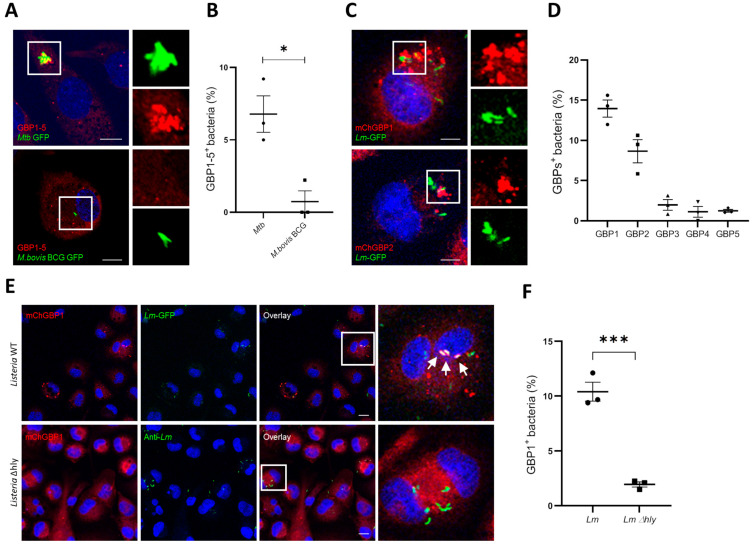
Colocalization of hGBPs with pathogens destabilizing phagosomal membranes. (**A**) The colocalization of hGBP1-5 with *Mycobacteria*. THP-1 cells were stimulated with IFN-γ overnight, and then infected with *Mycobacterium tuberculosis* GFP (Mtb-GFP) or *M.bovis* BCG GFP for 4 h, fixed and stained with the antibody against hGBP1-5. (**B**) Quantification of hGBP1-5 positive Mtb-GFP or *M. bovis* BCG GFP. (**C**) The colocalization of hGBP1 or hGBP2 with *Listeria monocytogenes* (Lm). A549 cells expressing mCherry-hGBP1 or hGBP2 were infected with *L. monocytogenes* GFP (Lm-GFP) for 1 h. (**D**) Quantification of the percentage of mCherry-GBP1/GBP2 colocalization with Lm. (**E**) hGBP1 association to Lm is dependent on listeriolysin O (LLO). A549 cells expressing mCherry-hGBP1 were infected with Lm-GFP or Lm Δhly for 1 h. Lm Δhly were then visualized with an antibody against Lm. DAPI was used for nuclear staining. Arrows indicate colocalization of mCherry-hGBP1 with Lm. Scale Bar: 20 µm. For each experiment, 25 fields at 20× magnification were captured and blindly analyzed. Data are shown as mean ± SEM of three independent experiments; *p*-values were calculated using Student’s *t*-test (**B**,**F**) or one-way ANOVA (D). (*) *p* < 0.05, (***) *p* < 0.001.

**Figure 3 ijms-24-09701-f003:**
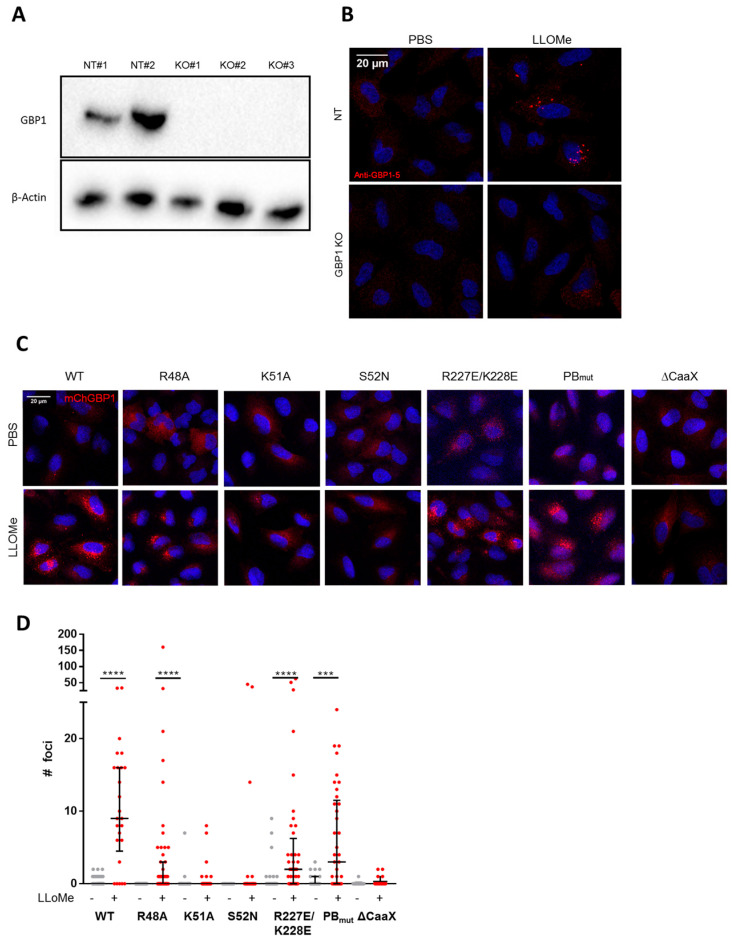
hGBP1 foci formation depends on GTP-binding and isoprenylation. (**A**) Western blot of hGBP1 in IFN-γ-primed A549 non-targeting (NT) cells (NT#1, NT#2) and GBP1 KO cell clones (KO#1, KO#2, KO#3). (**B**) Representative images of hGBPs foci formation in hGBP1 KO cells upon lysosomal damage. IFN-γ-primed A549 NT and hGBP1 KO cells were treated with PBS or LLOMe for 1 h and cells were then stained with antibody against hGBP1-5. (**C**,**D**) Representative images (**C**) and foci quantification (**D**) of A549 hGBP1 KO cells complemented with indicated mCherry-hGBP1 mutants, and then mock-treated or treated with LLOMe for 1 h. Representative images from three independent experiments are shown. Scale bar: 20 µm; *p*-values were calculated using one-way ANOVA with Kruskal–Wallis test for multiple comparisons. (***) *p* < 0.001, (****) *p* < 0.0001.

**Figure 4 ijms-24-09701-f004:**
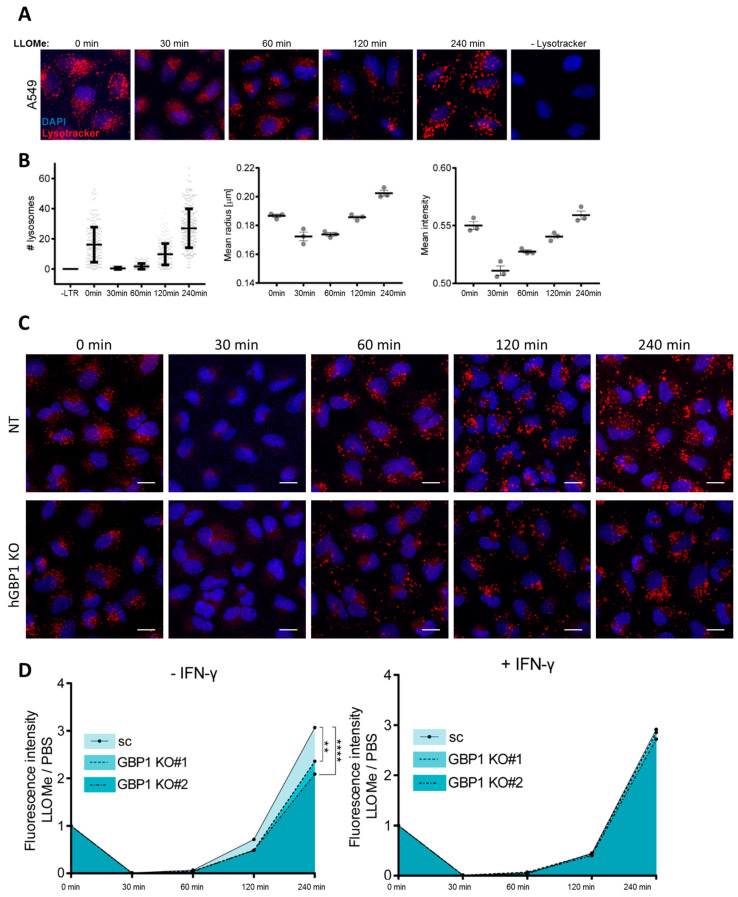
hGBP1 mediates the recovery of endolysosomal integrity post-damage. (**A**) Loss and recovery of lysosomal integrity over time following lysosomal damage. A549 cells were incubated with LysoTracker Red (LTR) 30 min prior to treatment of LLOMe at different time points. (**B**) Quantification of numbers, mean radius, and intensity of lysosomes. For the number of lysosomes per cell, each dot represents a cell. Means ± SD of three independent experiments. Mean radius and intensity represent means ± SEM of three independent experiments. Each dot represents one independent experiment. (**C**,**D**) Representative images (**C**) and quantification (**D**) of Lysotracker staining in A549 NT and hGBP1 KO cells upon lysosomal damage. (**C**,**D**) A549 NT and hGBP1 KO cells (KO#1 and KO#2) were treated with or without IFN-γ overnight, then cells were incubated with Lysotracker 30 min prior to mock or LLOMe treatment for an indicated time. Lysotracker intensity of LLOMe treatment is normalized to mock. Data represent the mean of relative intensity from three independent experiments; *p*-values were calculated using two-way ANOVA with Bonferroni’s multiple comparison test. (**) *p* < 0.01, (****) *p* < 0.0001.

**Figure 5 ijms-24-09701-f005:**
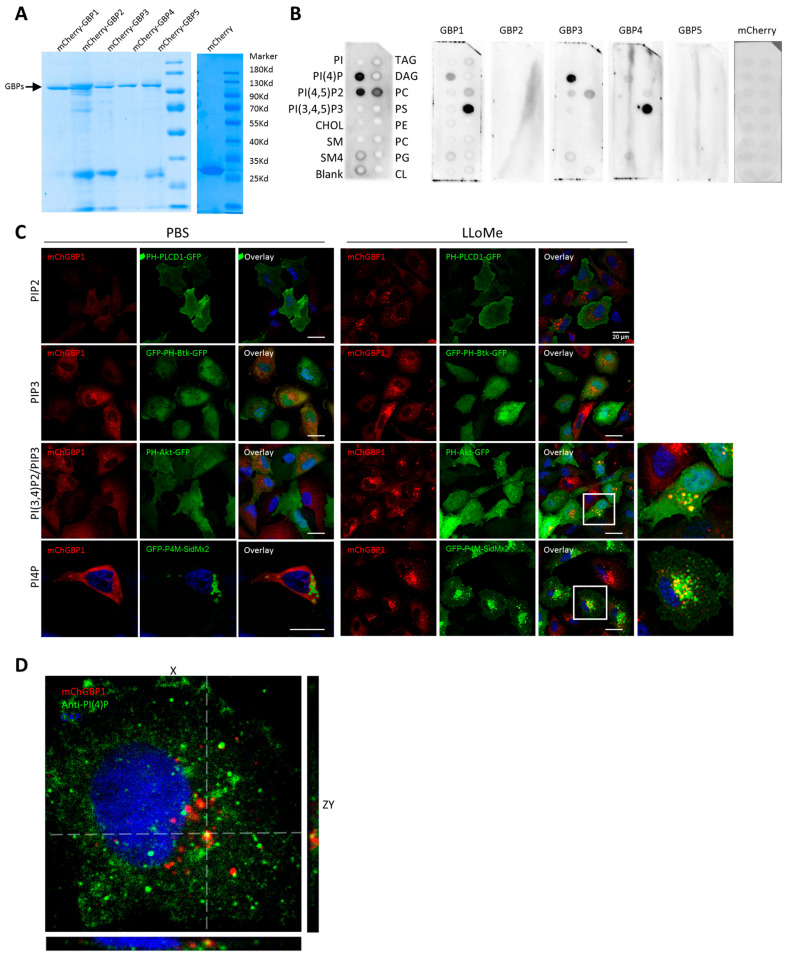
hGBP1 binds to PI4P upon endolysosomal damage. (**A**) SDS-PAGE analysis of purified mCherry and mCherry-hGBP1-5 proteins. (**B**) Binding of mCherry and mCherry-hGBPs to membrane lipids in vitro. GST-PLC-d1-PH was utilized as the positive control. (**C**) Colocalization analysis of hGBP1 with various GFP-tagged lipid sensors in cells upon endolysosomal damage. A549 cells expressing mCherry-hGBP1 were nucleofected with GFP-tagged lipid sensors for PI(4)P (GFP-P4M-SidMx2), PIP3/PI(3,4)P2 (PH-Akt-GFP), PIP3 (PH-Btk-GFP), and PIP2 (PH-PLCD1-GFP), then treated with LLOMe or mock (PBS) for 1 h. (**D**) Colocalization of hGBP1 with endogenous PI4P in cells upon lysosome damage. PI4P was visualized with a monoclonal IgM antibody (Echelon Biosciences, Salt Lake City, UT, USA). Scale bars: 20 µM. Results are representative of three independent experiments.

**Figure 6 ijms-24-09701-f006:**
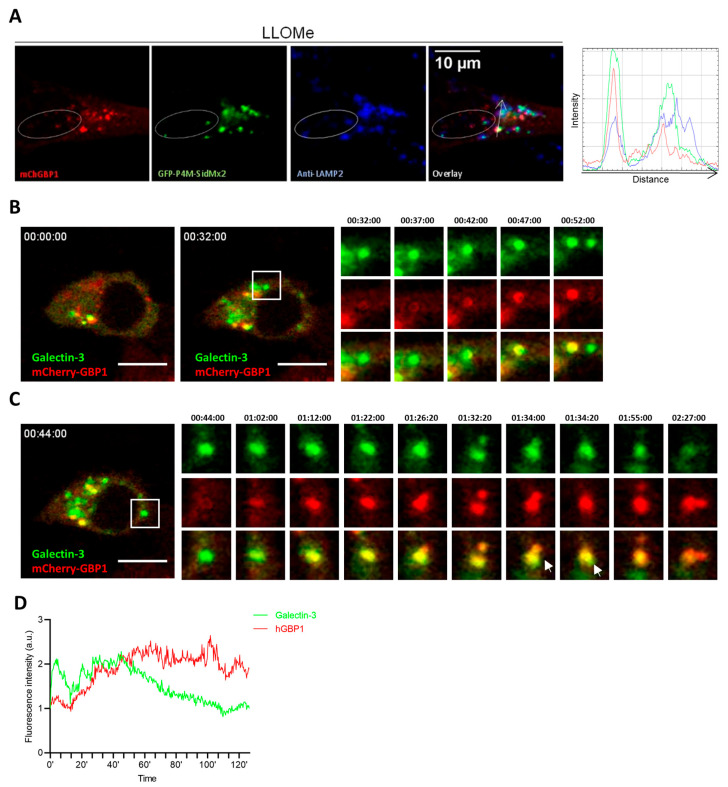
hGBP1 is recruited to damaged endolysosomes and mediates their recovery. (**A**) Colocalization of hGBP1 and PI4P on damaged lysosomes. A549 cells expressing mCherry-hGBP1 were nucleofected with GFP-tagged PI4P lipid sensor-GFP-P4M-SidMx2, treated with LLOMe for 1 h, and stained with antibody against LAMP2. Line profiles were performed with ImageJ, demonstrating fluorescence intensities from different channels along the arrow. Scale bar: 10 µm. (**B**,**C**) Live-cell imaging analysis of hGBP1 dynamics upon lysosomal damage. A549 cells expressing mCherry-hGBP1 were nucleofected with YFP-Galectin-3. Cells were treated with LLOMe (1 mM) and live imaging started 10 min post-treatment with image interval of 20 s. Arrows indicate fusion events. Scale bar: 20 µm. (**D**) Quantification of the fluorescence intensity of mCherry-hGBP1 and YFP-Galectin-3 on damaged endolysosome in (**C**). The intensity was normalized to the values at 42:00. Results are representative of two independent experiments.

**Table 1 ijms-24-09701-t001:** hGBPs primers used for cloning into the lentiviral vector.

hGBP1	Fw: AGTAGGATCCGCATCAGAGATCCACATGACAG
	Rv: TACTGCGGCCGCTTATTAGCTTATGGTACATGCCTTTCG
hGBP2	Fw: AGTAGGATCCGCTCCAGAGATCAACTTGCCG
	Rv: TACTGCGGCCGCTTATTAGAGTATGTTACATATTGGCTC
hGBP3	Fw: AGTAGGATCCGCTCCAGAGATCCACATGACAG
	Rv: TACTGCGGCCGCTTATTAGATCTTTAGCTTATGCGAC
hGBP4	Fw: AGTAGGATCCGGTGAGAGAACTCTTCACGCTG
	Rv: TACTGCGGCCGCTTATTAAATACGTGAGCCAAGATATTTTG
hGBP5	Fw: AGTAGGATCCGCTTTAGAGATCCACGCTTTAG
	Rv: TACTGCGGCCGCTTATTAGAGTAAAACACATGGATCATC

**Table 2 ijms-24-09701-t002:** Primers for hGBP1 mutagenesis.

hGBP1 R48A	Fw: GGGCCTCTACgcCACAGGCAAATC
	Rv: ACAATTGCCACCACCACC
hGBP1 K51A	Fw: CCGCACAGGCgcATCCTACCTG
	Rv: TAGAGGCCCACAATTGCC
hGBP1 S52N	Fw: CACAGGCAAAaaCTACCTGATGAACAAG
	Rv: CGGTAGAGGCCCACAATT
hGBP1 R227E/K228E	Fw: gcagcagcgGCATGTACCATAAGCTAATAAG
	Rv: tgccgctgcCGTCTGGAGATCCTGTATC
hGBP1 ΔCaaX	Fw: TAATAAGCGGCCGCACTC
	Rv: TGCCTTTCGTCGTCTCATTTTC
hGBP1 PBmut	Fw: ACTCTGTATCgaggaaTTCTTCCCAAAGAAAAAATG
	Rv: CTGGGCAGGTTAAAAGTTTC

## Data Availability

Not applicable.

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
