# Peer review of "Human GBP1 Is Involved in the Repair of Damaged Phagosomes/Endolysosomes"

_ijms, 2023, doi:10.3390/ijms24119701_

Round 1

Reviewer 1 Report

In this paper the authors use cell imaging to show the recruitment of GBPs during phagolysosomal damage during infection with M. tuberculosis and L. monocytogenes. The study is very well conducted with the use of relevant molecular tools. Overall the work clearly deserves to be published, but there are some suggestions that could improve the quality of the work.

Comments:

-       For figure 1, it would be really interesting to follow GBP recruitment in kinetics for Mtb and Lm macrophages infection. It is possible that the rupture is higher 24 hours post infection in the Mtb model in particular.

-       for figure 6, it would be very interesting to perform the experiment in an infectious context in order to follow the phenomenon. for example, is galectin 3 recruited to the phagosome of Mtb and Lm?

Minor comment:

-       It is unusual to consider M tuberculosis as a gram-positive bacterium.

Author Response

We highly appreciate the time and efforts that the reviewers dedicated to providing feedback on our manuscript and we are very grateful for the insightful comments. Please see the attachment for the point-by-point response.

Reviewer 2 Report

In the article “Human GBP1 is involved in the repair of damaged phagosomes and endolysosomes” by Buijze et al., the authors present how Guanylate-binding protein 1(GBP1) interacts with several phospholipids and is recruited to damaged endosomes. The article is well written and describes interesting observations. However, several questions should be addressed.

Major comments

1.   Figure 1: THP1 cells were infected with Mycobacteria in Figure 1A, A549 overexpressing mCherry-tagged GBPs were infected with Listeria in Figure 1C and 1E. These are two different cell lines infected with two different bacterial pathogens. Additional data should be included with THP1 cells infected with Listeria and A549 overexpressing mCherry-tagged GBPs infected with Mycobacteria.

For the microscopy data, endosomal markers (Rab5/Rab7/LAMP1) should be implemented to determine whether the bacteria are cytosolic or still present in an endosome.

Unless these experiments are performed, the conclusion (line 106-108) is not correct.

2.   Figure 2A: the authors should check for the identity of the perinuclear structures appearing after LLOME treatment of A549 transfected with mCherry-GBP1 by immunofluorescence labeling of endosomal and Golgi markers.

3.   Figure 2D: do A549 cells express hGBP1 upon IFN treatment? Western blot analysis of mock and IFN-treated A549 would be interesting. If A549 express endogenous hGBP1 upon IFN treatment, why use ectopic expression of mCherry-hGBP1?

4.   Figure 2F: the images do not show any mCherry-GBP foci, how were they counted in Figure 2G?

5.   Figure 3B: the antibody used is an anti-GBP1-5, why did the authors not use the anti-GBP1 antibody as in Figure 3A?

6.   Supplementary Figure 2A and line 193-194: the microscopy image does not show that “hGBP1 specifically colocalizes with LAMP2”, the GBP1 puncta has some LAMP2 staining but the LAMP2 staining remains mainly scattered in the cell.

7.   Figure 4A and 4C: The LTR intensity seems higher after 240 minutes of recovery, why is not this time point used/presented in Figure 4C?

8.   Figure 5B: a negative control is missing, as PIP strips were incubated with mCherry-GBPs-6xHis, a proper negative control would be PIP strips incubated with mCherry-6xHis.

9.     Figure 5C: The authors should probe the presence of PI(3,5)P2 (with ML1-N for example)on puncta formed by hGBP1 as this lipid is enriched on endolysosomes.

10.  Figure 6B and 6C: The experiment should be repeated with hGBP1 mutants described in Figure 3C in order to confirm the specificity of interaction and role of hGBP1 domains in its recruitment to damaged endosomes. Furthermore, the experiment does not show action of the autophagy machinery or repair of the endosome, thus it cannot be stated that hGBP1 participates in the repair of the endosome (Line 294). The same experiment in the presence of autophagy inhibitors and autophagy markers (LC3) should thus be performed.

11.  Statistics should be added to Figure 7. This one could be merged with Figure 1 or used as a supplementary figure as it does bring any insights into hGBP1 function at the level of bacterial infection control. Experiment in Figure 7C could be further investigated, as overexpression of hGBP2 and hGBP3 seem to facilitate infection at later time points.

Minor comments

References 40 (Line 665-666) and 50 (Line 688-689) are incomplete, please review all references.

Line 335: Toxoplasma and Salmonella should be in italic, please check thorough the manuscript.

Author Response

We highly appreciate the time and efforts that the reviewers dedicated to providing feedback on our manuscript and we are very grateful for the insightful comments. Please see the attachment of the point-by-point responses.

Round 2

Reviewer 2 Report

The authors made the required changes to the manuscript, references still need to be proofread (reference number 40 refers to "Science 360" instead of "Science 360(6384):eaar5078")

Author Response

Thank the reviewer for the careful reading. We have changed the reference style according to the suggestions of MDPI.